# WORLDWIDE FEDERATED TRAINING OF LANGUAGE MODELS

## ABSTRACT

Language Model (LM) training requires vast datasets, raising legal, ethical, and practical concerns. Federated learning (FL) offers an alternative by enabling organizations to collaboratively leverage untapped reserves while minimizing data movement. However, scaling FL globally introduces challenges such as restrictions on data movement, privacy, and statistical data heterogeneity. We propose *Worldwide Federated Language Model Training* (*WorldLM*), a system that builds **federations of federations**. *WorldLM* enables each federation to autonomously meet jurisdictional or competitive constraints while managing statistical heterogeneity through *attention-based aggregation* of key layers and *cross-federation information sharing* via residual embeddings. In terms of perplexity, *WorldLM* outperforms standard FL and other federated baselines by up to $1.91\times$ and $3.3\times$ respectively. *WorldLM* scales to models with 400M parameters, achieving $1.39\times$ lower perplexity than centralized counterparts while approaching the performance of perfectly localized models trained in an infinite-data regime. Additionally, under differential privacy constraints, *WorldLM* proves highly resilient in performance compared to standard FL methods, which diverge. These results establish *WorldLM* as an effective means for pre-training across geographic and legal boundaries.

## 1 INTRODUCTION

Language models (LMs) require extensive computational resources and vast amounts of curated text data, often centralized in data centers (Scao et al., 2022; Dubey et al., 2024). This centralized paradigm raises concerns (Bommasani et al., 2021) about data ownership (Council of European Union, 2021), privacy, and copyright issues (Grynbaum & Mac, 2023), as well as the limited availability of high-quality linguistic data (Villalobos et al., 2022). Federated Learning (FL) has emerged as a promising alternative that allows organizations to collaboratively train models without sharing raw data (McMahan et al., 2017; Douillard et al., 2023; Sani et al., 2024; Woisetschläger et al., 2024a). FL reduces the need for data movement and synchronization overheads (Rajbhandari et al., 2020; Zhao et al., 2023), while incorporating privacy-preserving techniques such as differential privacy (DP) (Wei et al., 2020) and secure aggregation (Bonawitz et al., 2016). However, scaling FL globally introduces challenges like federated governance and statistical heterogeneity.

The challenge of **federated governance** (González-García et al., 2021) arises when participants in a federated system operate under varying legal, privacy, and security constraints. For example, participants in the European Union must comply with the GDPR (European Parliament & Council of the European Union, 2016), which imposes rules on cross-border data sharing, while others may be free to operate without such constraints. The second challenge, **statistical heterogeneity**, occurs when different participants hold non-IID data. This heterogeneity can arise from differences in language (Conneau et al., 2020) or domain. This can lead to slower convergence rates and even divergence in standard FL settings (Zhou et al., 2023; Ye et al., 2024). Moreover, **dataset geography** (Faisal et al., 2022) often exacerbates these issues, as data collected from different regions may exhibit inherent clustering, requiring careful optimization across global and local distributions.

To address these challenges, we propose the *Worldwide Federated Language Model Training* (*WorldLM*) system, built on the idea of **federations of federations**. This hierarchical approach allows federations to collaborate while maintaining local autonomy to respect regulatory and competitive constraints. *WorldLM* brings the following innovations:

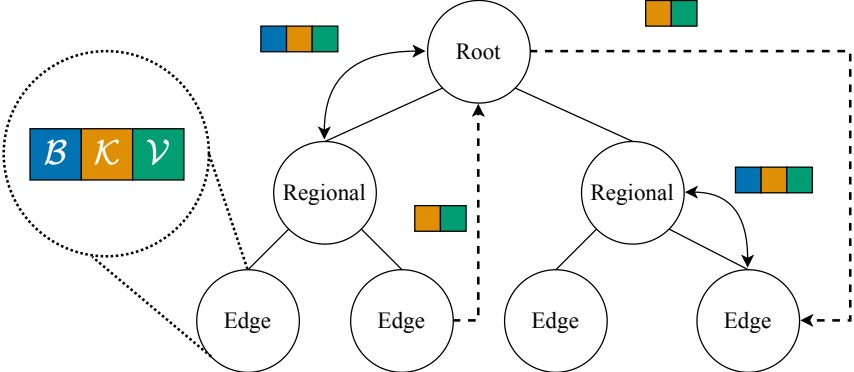

Figure 1: Overview of **WorldLM**'s hierarchical structure. Federations exchange backbone and personalized key layers, while residual embeddings are selectively routed between parent and child nodes to facilitate cross-federation information sharing.

1. **Arbitrary Federation Structure:** *WorldLM* allows federations to account for their unique legal, privacy, and security constraints while participating in a global training process. Each federation can operate under separate legal frameworks, while leveraging privacy-enhancing techniques like differential privacy (Wei et al., 2020) and secure aggregation (Bonawitz et al., 2016). This provides a flexible structure that enables seamless collaboration across jurisdictions and industries without compromising compliance.

2. **Partially-Personalized Aggregation:** To address the statistical heterogeneity inherent in real-world federated systems, *WorldLM* incorporates both a shared model backbone and personalized key layers specific to each node. These key layers are aggregated using an attention-based mechanism that balances global and local objectives. Results on multilingual pre-training show that *WorldLM* achieves a perplexity $1.91\times$ lower than standard FL, $1.86\times$ lower than Hierarchical Federated Averaging (*HierFAVG*) (Liu et al., 2020) and $3.3\times$ lower than Federated Learning With Personalization Layers (*FedPer*) (Arivazhagan et al., 2019; Li et al., 2022).

3. **Cross-Federation Information Sharing:** In cases where participants' data is more similar to data in other sub-federations, *WorldLM* allows for the sharing of *residual* layers, enabling transformer layers badly fit to the local distribution to be routed to more relevant federations. This cross-federation information sharing mechanism ensures that participants whose data is dissimilar from that of their peers can still benefit from global collaboration, while minimizing communication overhead by sending only selected residual blocks.

We show that *WorldLM* scales effectively to larger models by training a model with 400M parameters, the same size as the 2023 SOTA (Douillard et al., 2023) for federated pre-training, on the *MC4* dataset, achieving a $1.39\times$ lower perplexity compared to centralized baselines. We also evaluate *WorldLM* on a gauntlet of 35 downstream tasks organized in 5 categories and show average improvements of $4.49\%$ over standard FL on MC4. Thus, *WorldLM* provides an effective solution for training LMs across legal, socioeconomic, and cultural boundaries, diversifying the pool of available data away from its current geographic (Faisal et al., 2022) concentration.

## 2 BACKGROUND

The rise of large language models (LLMs), driven by established performance scaling laws (Hoffmann et al., 2022), has led to remarkable advancements in various downstream applications (Hu et al., 2021; Gema et al., 2023). Despite these advancements, pre-training remains reliant on large centralized datacenters due to high-bandwidth communication requirements, primarily driven by the synchronization overhead found in Ring AllReduce (Sergeev & Balso, 2018) used in standard training algorithms like FSDP (Rajbhandari et al., 2020; Zhao et al., 2023).

Recent work in federated learning (FL) has aimed to alleviate these communication bottlenecks. For example, Douillard et al. (2023) used Local SGD(Stich, 2019) to reduce synchronization. Following this, Sani et al. (2024) demonstrated that generative pre-training can be extended to FL, enabling models to leverage geographically distributed data while maintaining privacy. For scenarios where high-quality public datasets may become scarce (Villalobos et al., 2022), FL provides a novel means of tapping into previously inaccessible datasets held by different organizations or even across countries (OpenAI, 2023; Patel & Palazzolo, 2024). These developments prompt us to explore the global scale of federated learning, presenting new system and statistical challenges.

## 2.1 GLOBAL FEDERATED SYSTEMS

FL typically operates through a sequence of broadcast updates, local optimization, and aggregation processes (McMahan et al., 2017). This approach aligns with privacy regulations, particularly by minimizing the exchange of raw data (White House, 2013), making FL a suitable technique for compliance with frameworks such as the GDPR (European Parliament & Council of the European Union, 2016) and EU AI ACT (Council of European Union, 2021). However, global-scale FL introduces two primary challenges: **statistical heterogeneity** and **system heterogeneity**.

**Statistical heterogeneity** emerges when participating clients possess vastly different datasets, such as varying languages, regions, or domains, each with differing statistical properties (Zhou et al., 2023; Kairouz et al., 2021). When naive aggregation methods, such as those used in **FedAvg**, are applied across highly non-IID client datasets, the resulting model updates may conflict, leading to slower convergence or even performance degradation (Charles et al., 2021). This is analogous to the generalization gap seen in large-batch training (Keskar et al., 2017). To address statistical heterogeneity, better aggregation and personalization methods are needed.

**System heterogeneity** arises because clients within federated systems often operate under diverse conditions, including varying computational power, network bandwidth, and availability. These discrepancies can create "stragglers" that slow down the training process as the central server waits for slow or resource-constrained clients (Huba et al., 2022; Bonawitz et al., 2019). Such disruption delays convergence and increases the risk of uneven model updates.

Another critical aspect is **federated governance**, which governs how data can be shared and models collaboratively trained across legal and regulatory boundaries. In a cross-border context, Federated learning must respect local privacy laws, such as GDPR or the EU AI ACT, which dictate the constraints on data sharing across countries (European Parliament & Council of the European Union, 2016) and focus on mitigating biased outcomes (Council of European Union, 2021, Art. 10.2f,fa). Privacy-enhancing technologies (PETs) such as differential privacy (DP) (Wei et al., 2020) and secure aggregation (Bonawitz et al., 2016) may ensure the secure collaboration of clients despite differing privacy regulations. Further details on the legal implications of FL are available in Appendix A.5.

## 2.2 RELATED WORK

Personalized Federated Learning (PFL) (Tan et al., 2021) aims to enhance performance on a given client's data. One means of achieving this is creating hybrid models wherein common layers are shared while specific layers are customized. For instance, Li et al. (2022) and Arivazhagan et al. (2019) propose FL With Personalization Layers (*FedPer*) a method that personalizes deeper layers while sharing shallower ones, mitigating heterogeneity at the cost of decreased information sharing.

Client clustering methods (Sattler et al., 2021; Briggs et al., 2020) attempt to group clients based on model similarity but avoid considering the structural intricacies of neural networks, often reducing model embeddings to insufficient single-scalar compatibility values. Together with common concerns for clustering hyperparameters, this limits the practical usability of such methods.

Existing hierarchical systems like Liu et al. (2020) and Luo et al. (2020) focus primarily on communication efficiency by grouping clients under edge servers and performing aggregation via a simple extension of FedAvg (McMahan et al., 2017), called Hierarchical Federated Averaging(*HierFAVG*). However, they do not factor in **data heterogeneity** and aim to create one global model. More recent hierarchical methods, such as those in Mhaisen et al. (2022), integrate data heterogeneity into edge-server assignments but still rely on FedSGD, which is known to converge slower than FedAvg (McMahan et al., 2017).

# 3 WORLDWIDE FEDERATED TRAINING OF LANGUAGE MODELS (*WorldLM*)

Given the interlocking nature of legal, privacy, and security concerns, we assume organizations find collaborating with enterprises operating in the same geographic region, legal jurisdiction, or industry easier. Facilitating collaboration in edge-device FL (Kairouz et al., 2021) requires a fully decentralized collaborative learning paradigm (Kairouz et al., 2021; Zantedeschi et al., 2020) with a graph encoding client compatibility. For training LMs, where the participating entities are stable organizations, we argue that a **federation of federations** approach, as portrayed in Fig. 1, provides an attractive compromise between standard federated learning and decentralized learning. WorldLM employs a custom aggregation procedure and information-sharing mechanism to optimize for a given hierarchical data distribution.

## 3.1 HIERARCHICAL DATA DISTRIBUTIONS

We recursively define the data distribution of a federation based on the data distribution of its constituent sub-federations. Intra-federation heterogeneity pertains to the heterogeneity across children in the same sub-federation, while inter-federation heterogeneity pertains to the heterogeneity between any two federations that do not share the same parent. The degree of heterogeneity can be measured using common distance metrics between distributions, such as Earth Mover's Distance (Zhao et al., 2018). Under this definition, an "empty" hierarchical federation contains no members, whereas a "trivial" hierarchical federation consists of one member.

Formally, for a sub-federation $Q$ with root server $q$, we define its data distribution as $\Omega_q = \cup_{c \in C_q} \Omega_c \cup \omega_q$, which is a mixture of the heterogeneous data distributions $\{\Omega_c\}_{c \in C_q}$ of the set of children $C_q$ and the root server data $\omega_q$. Similarly, for another sub-federation $P$ with root server $p$, root server data $\omega_p$, data distribution $\Omega_p = \cup_{c \in C_p} \Omega_c \cup \omega_p$, and a set of children $C_p$ with data distributions $\{\Omega_c\}_{c \in C_p}$, the intra-federation heterogeneity is defined as the degree of heterogeneity across $\{\Omega_c\}_{c \in C_q}$ and $\{\Omega_c\}_{c \in C_p}$. The inter-federation heterogeneity is determined between $\Omega_p$ and $\Omega_q$.

A practical example: if $\{\Omega_c\}_{c \in C_q} \sim \text{LDA}_{0.1}$ and $\{\Omega_c\}_{c \in C_p} \sim \text{LDA}_{1000}$, then the intra-federation heterogeneity of $Q$ is greater than that of $P$.

## 3.2 PARTIALLY-PERSONALIZED AGGREGATION

Federated learning can train powerful feature extractors beneficial to all WorldLM participants due to its meta-learning properties (Nichol et al., 2018; Fallah et al., 2020; Lee et al., 2023). However, participants in federated training may hold heterogeneous data, such as different languages or domains (e.g., news versus scientific publications). Thus, the feature extractor needs adaptation for each actor and sub-federation, necessitating a departure from the standard FL objective. WorldLM, inspired by split-learning and personalized techniques (Arivazhagan et al., 2019; Li et al., 2022), partitions the model into a backbone $\mathcal{B}$, comprising the majority of the model's parameters, and a set of partially personalized key layers $\mathcal{K}$ specific to each node.

The backbone parameters $\mathcal{B}$ are trained using FL aggregation algorithms such as FedAvg (McMahan et al., 2017) or FedOPT (Reddi et al., 2021). Algorithm 1 outlines our method, with key procedures distinguished by different colors and independent numbering per color. As shown in Algorithm 1 (L.1), for each round $k$, the server broadcasts $\mathcal{B}$ to its children $c \in C_q$. When a child begins execution, it loads or initializes its model ( L.2 ), replaces its previous backbone with the received one ( L.5 ), and aggregates its key layers with those of the parent ( L.6 ). This aggregation is performed on a per-key-layer basis ( L.2, 4 ) using an attention mechanism ( L.5 ) with average pooling ( L.6 ). The node then executes local training (L.4) followed by residual routing (L.6). Residual routing sends residual layers received from the parent ( L.3 ) to the child with the highest similarity ( L.4 ). If the child is a leaf node (a trivial federation), the residual is aggregated; otherwise, it is routed further. The node then recursively executes its descendants (L.7 − 8) and computes pseudo-gradients $\Delta_c^t$. These pseudo-gradients are aggregated and applied to the server's backbone via ServerOpt (L.11). The model aggregates key layers of its descendants by repeating the parent-to-child aggregation procedure for each child (L.14) followed by average pooling. Finally,

---

**Algorithm 1** WorldLM Federation Execution Algorithm

---

**Require:** Node id $q$, set of descendants $C_q$, number of rounds $T_q$, parent backbone $\mathcal{B}_p$, key layer sequence $\mathcal{K}_p$
**Require:** Downstream residuals for aggregation $\mathcal{D}_a$, routing $\mathcal{D}_r^0$, LoadModel, ClientOpt, ServerOpt
**Require:** Similarity function $s : \mathbb{R}^n \times \mathbb{R}^n \to \mathbb{R}$ (default: cosine similarity)

---

1: **procedure** WORLDLMFIT($q, \mathcal{B}_p, \mathcal{K}_p, D_a, D_r^0$)
2:  $B^0, K^0, U^0 \leftarrow$ AGGPARENT($q, B_p, K_p, D_a$)
3:  **for** round $t \leftarrow 0, \ldots, T_q - 1$ **do**
4:   $\mathcal{B}^t, \mathcal{K}^t \leftarrow$ ClientOpt($q, \mathcal{B}^t, \mathcal{K}^t$)                              ▷ Local optimization
5:   **if** $C_q \neq \emptyset$ **then**
6:    $A^t, R^t \leftarrow$ ROUTERESIDUALS($q, \mathcal{D}_r^t, C_q$)                       ▷ Route residuals
7:    **for** child $c \in C_q$ **do**                                             ▷ Train child nodes
8:     $\mathcal{B}_c^t, \mathcal{K}_c^t, \mathcal{U}_c^t \leftarrow$ WORLDLMFIT($c, \mathcal{B}^t, \mathcal{K}^t, A_c^t, R_c^t$ )        ▷ Recurse to child $c$
9:     $\Delta_c^t \leftarrow \mathcal{B}_c^t - \mathcal{B}^t$                                       ▷ Compute pseudo-gradient
10:    $\Delta^t \leftarrow \frac{1}{|C_q|} \sum_{c \in C_q} \Delta_c^t$                         ▷ Aggregate pseudo-gradients
11:    $\mathcal{B}^{t+1} \leftarrow$ ServerOpt($\mathcal{B}^t, -\Delta^t, t$)                       ▷ Update backbone
12:    $\mathcal{P} \leftarrow [\mathcal{K}_0^t, \ldots, \mathcal{K}_{|C_q|}^t]$                              ▷ Collect key layers
13:    $\mathcal{Q}, \mathcal{K}, \mathcal{V} \leftarrow \mathcal{P}, \mathcal{P}, \mathcal{P}$                           ▷ Set Q, K, V values
14:    $\mathcal{K}^{t+1} \leftarrow \sum_{i \in |\mathcal{Q}|} \frac{1}{|\mathcal{Q}|}$LAYERATTN($\mathcal{Q}_i, \mathcal{K}, \mathcal{V}$)            ▷ Apply attention
15:    $\mathcal{U}^{t+1}, \mathcal{D}_r^{t+1} \leftarrow$ PARTRESIDUALS($q, \mathcal{K}^{t+1}, \mathcal{V}, \mathcal{U}^t, \mathcal{D}_r^t$)      ▷ Partition residuals
16:  **return** $\mathcal{B}^{T_q}, \mathcal{K}^{T_q}, \mathcal{U}^{T_q}$                                         ▷ Return final state

---

1: **procedure** AGGPARENT($q, \mathcal{B}_p, \mathcal{K}_p, \mathcal{D}_a$)
2:  $\mathcal{U}^0, (\mathcal{B}^0, \mathcal{K}^0) \leftarrow \emptyset$, LoadModel($q$)
3:  **if** $q \neq 0$ **then**
4:   $\mathcal{K}, \mathcal{V} \leftarrow [\mathcal{K}^0, \mathcal{K}_p, $Expand($|\mathcal{K}^0|, \mathcal{D}_a$)]
5:   $\mathcal{B}^0 \leftarrow \mathcal{B}_p$
6:   $\mathcal{K}^0 \leftarrow$ LAYERATTN($\mathcal{K}^0, \mathcal{K}, \mathcal{V}$)
7:  **return** $B^0, \mathcal{K}^0, \mathcal{U}^0$

1: **procedure** LAYERATTN($\mathcal{Q}, \mathcal{K}, \mathcal{V}$)
2:  **for** $l \in |\mathcal{Q}|$ **do**
3:   $\mathcal{Z}_l \leftarrow 0$
4:   **for** $i \in |\mathcal{K}|$ **do**
5:    $\alpha_{i,l} \leftarrow \sigma(s(\mathcal{Q}_l, \mathcal{K}_{i,l}))$
6:    $\mathcal{Z}_l \leftarrow \mathcal{Z}_l + \alpha_{i,l} \mathcal{V}_{i,l}$
7:  **return** $\mathcal{Z}$

1: **procedure** ROUTERESIDUALS($q, D_r, C$)
2:  $A_c, R_c \leftarrow \emptyset, \emptyset, \forall c \in C$
3:  **for** $(v, l) \in D_r$ **do**
4:   $d \leftarrow \text{argmax}_{c \in C} s(v, \mathcal{K}_{c,l})$
5:   **if** IsLeaf($d$) **then**
6:    $A_d \leftarrow A_d \cup \{(v, l)\}$
7:   **else**
8:    $R_d \leftarrow R_d \cup \{(v, l)\}$
9:  **return** $A, R$

1: **procedure** PARTRESIDUALS($q, \mathcal{K}, \mathcal{V}, \mathcal{U}, \mathcal{D}_r$)
2:  **for** $l \in |\mathcal{K}|$ **do**
3:   $I^0 \leftarrow \{0, \ldots, |V| - 1\}$
4:   **for** $n \in \nu_{\mathcal{K}}$ **do**
5:    $m \leftarrow \text{argmin}_{c \in I^n} s(\mathcal{K}_l, \mathcal{V}_{c,l})$
6:    $I^{n+1} \leftarrow I^n \setminus \{m\}$
7:    $\mathcal{U} \leftarrow \mathcal{U} \cup \{(\mathcal{V}_{m,l}, l)\}$
8:  **if** $q = 0$ **then return** $\emptyset, \mathcal{D}_r \cup \mathcal{U}$
9:  **return** $\mathcal{U}, \mathcal{D}_r$

---

for each key layer ( L.2 ), the model selects the top-$\nu_{\mathcal{K}}$ ( L.4 ) most dissimilar child key layers. If the node is the root, it routes them down to the most relevant children ( L.8 ); otherwise, it passes the residuals to its parent ( L.9 ).

Local training ( L.4 ) can be executed either in-parallel with the node model treated as a client or sequentially. It is crucial to distinguish sequential from parallel steps, since the latter are averaged across clients, similarly to the gradients produced by one batch in standard SGD. Our work adopts the sequential approach, where the root executes by itself at the first level, while all leaves execute in parallel. For more details on how this impacts training, see Appendix A.1.

Unlike traditional personalization approaches, WorldLM optimizes the key layers $\mathcal{K}$ by considering models of both parent and children, aggregating each layer with an attention mechanism (Vaswani et al., 2017; Ji et al., 2019). The parameters of each layer across federation nodes serve as queries, keys, and values. To exploit data locality and node similarity within the same sub-federation, the attention mechanism is applied within a local context, either relative to the node's parent or children. This procedure allows WorldLM to manage statistical heterogeneity effectively when sub-federation nodes have similar data distributions. Unlike FedPer, which personalizes specific

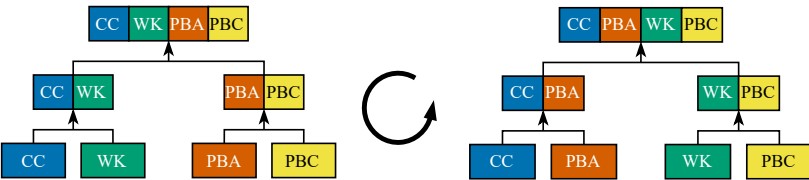

Figure 2: Data-perspective upon a hierarchical dataset constructed from `The Pile` (Gao et al., 2021). The LHS contains two naturally heterogeneous and quantity-skewed groupings of data sources, corresponding to organizations accessing data from the internet or the medical domain. We construct such groupings using the internet-based Common Crawl (`CC`) and Wikipedia (`WK`) versus the medical data of PubMed Abstracts (`PBA`) and PubMed Central (`PBC`). To test the effectiveness of *WorldLM* when such a cluster relationship is absent, we swap the position of the two smaller datasets.

layers without sharing information, WorldLM utilizes a hierarchical structure for better management of heterogeneity.

To determine the number of key layers, we followed empirical results from the literature on Federated Learning with Personalization layers and transfer learning. For example, the original work on BERT (Devlin et al., 2019) suggests concatenating the last four layers of a 12-layer model provides optimal transfer learning performance, while in federated contexts (Li et al., 2022; Arivazhagan et al., 2019), personalizing the last four layers is most effective. Given the identical structure of decoder-only transformers' blocks, we chose to use 30% of the blocks on average.

The attention-based aggregation of a node's children $\big(L.14\big)$ reduces to simple unweighted averaging if no cluster relationship exists in a sub-federation. In contrast, aggregation between a node and its parent $\big(L.2\big)$ focuses almost exclusively on the node's keys, approaching a personalization-layer strategy. Similarly, if a node's data distribution significantly differs from its peers, its key layers will be highly dissimilar and thus ignored in aggregation. A full description of the detailed mathematical logic of the algorithm is presented in Appendix A.3.

## 4 EXPERIMENTAL DESIGN

Given the recent emergence of federated generative pre-training and the lack of benchmark datasets, we evaluate *WorldLM* on tasks that approximate realistic scenarios for its application: (a) organizations in different industries collaborating to train an LM despite holding different genres of text, and (b) organizations trying to train an LM despite holding data in different languages.

### 4.1 FEDERATION CONSTRUCTION

To simulate organizations holding different genres of text, we partition `The Pile` (Gao et al., 2021) into its constituent heterogeneous datasets and construct federations of federations by bottom-up building different mixtures of datasets. As seen in Fig. 2, if the children of a node hold data from the Pile Common Crawl (`CC`), Wikipedia (`WK`), PubMed Central (`PBC`), and PubMed Abstracts (`PBA`) datasets, then the node itself will hold data proportional to their size. For this dataset, we use the common *gpt-neox-20b* English tokenizer also adopted by Sani et al. (2024).

For simulating geographically distributed systems, we use a subset of the Multilingual Colossal Common Crawl (`MC4`)(Xue et al., 2021), covering high and low-resource languages (Magueresse et al., 2020). Sub-federations are constructed based on language families similarly to `The Pile`. Given the larger vocabulary size (Xue et al., 2021) and consequent model size, we use a single federated structure partitioned with high-resource French (`FR`) and Italian (`IT`) on one side and lower-resource Bulgarian (`BG`) and Ukrainian (`UK`) on the other. For IID experiments, we use the standard Cleaned Colossal Common Crawl (`C4`) English dataset, partitioned into equal-sized shards using the same tokenizer as `The Pile`. Given the larger size of multilingual models, and the restricted heterogeneity of `The Pile`, we use `The Pile` for ablation studies while using `MC4` for crucial baselines comparisons.

Each federation in our hierarchy represents a substantial data holder, organized in a tree structure. Each node $q$ in the tree can have a set of children $C_q$. Trivial federations are represented by leaf nodes with no children ($C_q = \emptyset$). In this experimental setup, the root federation consists of seven participating nodes counting the root, then each regional federation contains three participating nodes, and each leaf is a federation of one. The configuration is reported in Fig. 2 while partitions and their sizes are detailed in Appendix A.2.

## 4.2 TASKS

For all experiments, we use decoder-only transformers (Brown et al., 2020) for the language modeling task. Given that we are concerned with partially personalized FL, we compare the personalized **local** perplexity of *WorldLM* against three relevant baselines representing alternative scenarios. First, we compare against standard FL with momentum (Huo et al., 2020; Douillard et al., 2023; Sani et al., 2024), which has no hierarchical structure and is challenging to integrate across heterogeneous participants. Second, we compare against Hierarchical FedAvg (*HierFAVG*)( (Liu et al., 2020), which does not have any personalization. Third, we compare against *FedPER* (Arivazhagan et al., 2019) which has personalized layers but no hierarchical structure. Finally, we compare against centralized training of a global model on all node data pooled together, which represents the standard pre-training recipe when all data can be centralized.

In addition to these baselines, we are also interested in evaluating upper bound of local performance on a given data distribution for a given model size, in order to asses how close *WorldLM* gets to it. For this purpose, we train models of the same size as *WorldLM* using an infinite-data regime for the given node. We achieve this by training models that are far too small compared to the size of the local dataset recommended by scaling laws (Hoffmann et al., 2022), specifically we use datasets appropriate for billion-scale models to complete one epoch while training 75M-400M models. These should be seen as idealized cases for what the participants could train if they had unbounded data, in which case they would not need to use federated training to begin with. For the rest of this work we shall use the term "infinite-data local models".

**Language Modeling:** We utilize models of four sizes: 75M, 125M, and 250M and 400M. Our federated training uses the parameters shown in Table 5, while local parameters are available in Table 6. We compare models that have executed for a given number of **sequential** steps. The step-wise execution in *WorldLM* (Algorithm 1) implies that each level of the hierarchy (Fig. 1) trains in parallel, while the levels execute sequentially in **three** stages: Root, Regional, and Edge. For the 400M, we only had the resources for comparison against centralized, as standard FL is much less computationally efficient than *WorldLM* due to not having this three-stage execution (Appendix A.1.2).

For the local training of each node, samples are drawn proportionally to the data source sizes to ensure balanced and representative batches. Each node's dataset mixture is constituted by combining streams from relevant data sources while employing balanced sampling with shuffling. Since each node performs the same number of local steps, the size of a data source is primarily relevant in determining sampling ratios locally. Full details on the impact of sampling are available in Appendix A.1.

**Privacy and Security:** To validate the effectiveness of *WorldLM* in enhancing privacy and security, we simulate differentially private training, where the leaf nodes of a hierarchy contain potentially sensitive information and use *DPFedAvg* (Wei et al., 2020; Andrew et al., 2021) instead of standard averaging. This requires gradient clipping and the injection of Gaussian noise. For this, we assume that the `Pile-CC` and `WK` leaf clients of `The Pile` require DP, injecting noise with $\sigma = 0.5$ and clipping gradients to the median $l_2$ norm of the previous round (using a bound of $1.0$ for the first round as in centralized ML (Scao et al., 2022)).

## 5 EVALUATION

Our results for the language modeling task (Tables 1 and 2 and Figs. 3 and 5) show that *WorldLM* offers the desired compromise between global performance and local personalization.

Table 1: Language modeling personalized performance (over **local** client test sets) of *WorldLM* on `MC4` in terms of perplexity (the lower, the better). We compare against standard FL, infinite-data fully local models, `HierFAVG`, `FedPer` and centralized models. The latter is trained on the union of all local training sets. *WorldLM* outperforms standard FL, reaching a perplexity $1.91\times$ lower for the 250M. Furthermore, while it almost reaches the performance of centralized training for the 250M model, it outperforms it for the 400M, reaching a perplexity $\mathbf{1.39\times}$ lower.

| | Collaborative | | | | Non-Collaborative | |
| --- | --- | --- | --- | --- | --- | --- |
| **Model** | *WorldLM* | **FL** | **HierFAVG** | **FedPer** | **Local** | **Centralized** |
| **250M** | $\mathbf{80.47 \pm 68.53}$ | $153.27 \pm 95.47$ | $149.00 \pm 95.55$ | $269.85 \pm 129.00$ | $\mathbf{45.47 \pm 31.13}$ | $72.21 \pm 49.78$ |
| **400M** | $\mathbf{44.46 \pm 29.25}$ | - | - | - | - | $62.08 \pm 35.27$ |

## 5.1 *WorldLM* Outperforms FL and Centralized On Non-IID Data

Table 2 and Table 1 show that *WorldLM* is capable of outperforming standard FL in terms of **local** personalized performance for datasets that exhibit statistical heterogeneity while obtaining similar performance on the IID `C4` dataset.

*WorldLM* approaches the performance of a fully centralized training paradigm for the 75M and 125M models on `The Pile`, and sometimes exceeds it for the 250M and 400M models on `MC4`. For the 250M model trained on `MC4`, *WorldLM* reaches a perplexity $\mathbf{1.91\times}$ lower. For for the less heterogeneous monolingual `The Pile` it bring improvements of $1.15 \times -1.45\times$. The episodic nature of *WorldLM* 's execution, characterized by periodic increases in train perplexity (Fig. 3) when revisiting different stages, indicating its adaptive strategy in the aggregation process. Each subsequent revisiting of a stage results in a lower starting and ending perplexity, as the model learns to simultaneously optimize for the hierarchical data distribution of the entire tree and for the local distribution.

For the 250M model trained on `MC4`, *WorldLM* also show substantial improvements over *FedPer* and *HierFAVG*. Specifically, *WorldLM* achieved a $\mathbf{1.86x}$ lower perplexity than Hierarchical Federated Averaging (*HierFAVG*) by accounting for both inter- and intra-federation heterogeneity more effectively. When comparing against a hierarchical version of Federated Learning with personalization layers (*FedPer*), *WorldLM* showed a $\mathbf{3.3}\times$ lower average validation perplexity. Crucially, this improvement is due to the effective incorporation of relevant information from multiple participants across hierarchical levels, which is not feasible for *FedPer*.

The $400M$ model further exemplifies the scalability and robustness of *WorldLM*, demonstrating a **significant improvement** over its centralized counterpart, reaching a perplexity $\mathbf{1.39\times}$ lower.

## 5.2 *WorldLM* Is Robust To DP And Alternative Hierarchies

We evaluate the robustness of *WorldLM* to differential privacy and to data heterogeneity. For the first, Table 3 shows that *WorldLM* is generally more robust than standard FL to the gradient clipping and noise that DP injects into the models of two leaf nodes. Standard FL diverges immediately due to its inability to suppress the impact of DP on the global model. The personalized keys of *WorldLM*, on the other hand, can ignore the impact of the noise entirely, as seen in Fig. 4. Furthermore, the additional per-level momentum mechanism of *WorldLM* allows it to stabilize the backbone training.

Table 2: Language modeling personalized performance (over **local** client test sets) of *WorldLM* in terms of perplexity (the lower, the better). We compare against standard FL, infinite-data fully local, and centralized models. The latter is trained on the union of local training sets. *WorldLM* outperforms standard FL across Non-IID English dataset partitions, reducing perplexity by $1.15 \times -1.45\times$.

| | | Collaborative | | Non-Collaborative | |
| --- | --- | --- | --- | --- | --- |
| **Dataset** | **Model** | *WorldLM* | **FL** | **Local** | **Centralized** |
| **Pile** | **75M** | $\mathbf{73.82 \pm 44.18}$ | $107.31 \pm 52.50$ | $\mathbf{40.66 \pm 25.28}$ | $85.81 \pm 24.42$ |
| **Pile** | **125M** | $\mathbf{48.34 \pm 32.41}$ | $53.92 \pm 24.24$ | $\mathbf{24.83 \pm 12.47}$ | $29.61 \pm 13.17$ |
| **C4** | **75M** | $167.31 \pm 2.92$ | $\mathbf{145.32 \pm 3.53}$ | N/A | $67.01 \pm 1.67$ |

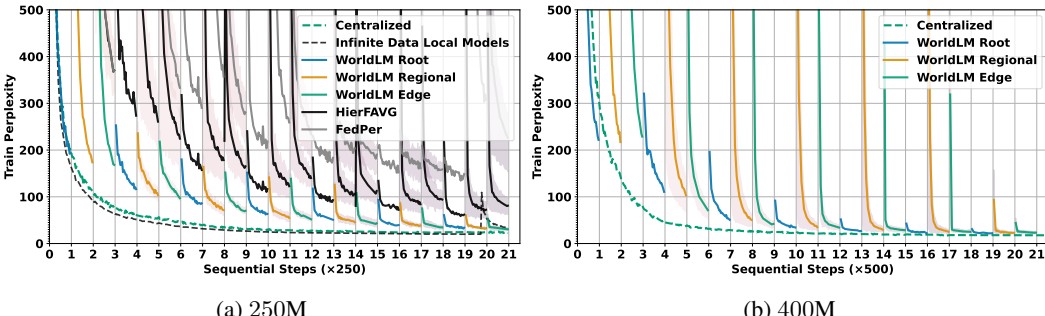

|           | (a) 250M | (b) 400M |

Figure 3: *WorldLM* performance of the multilingual models trained on a heterogeneous partitioning of `MC4` constructed analogously to Fig. 2 using `IT` and `FR` on one side and `UK` and `BG` on the other. While standard FL stops improving after round 15, *WorldLM* reaches a performance close to the centralized model and even the infinite-data local models. Furthermore, for the 250M bmodel it outperforms *HierFAVG* and *FedPer* for the entire traininig. For the 400M model, which matches the 2023 SOTA for federated pre-training, we compare against centralized only due to limited compute and show comparable performance as early as round 15 with a better final average perplexity (Table 1)

Table 3: Results evaluating privacy and robustness. *WorldLM* is highly resilient to DP with $\sigma = 0.5$ being applied over two of its leaf participants due to its aggregation procedure. It is also capable of handling the alternative arrangement of `WK` and `PBA`, only losing slightly in val perplexity to FL.

| Method | Pile | $DP_{CC,WK}$ | $DP_{PBC,PBA}$ | Pile (A) |
|---|---|---|---|---|
| *WorldLM* | **73.82 ± 44.18** | **101.78 ± 88.48** | **103.68 ± 90.53** | 140.05 ± 100.52 |
| FL | 107.31 ± 52.50 | 724.56 ± 251.89 | 724.24 ± 250.98 | **107.31 ± 52.50** |

This indicates that additional means of accounting for noise may be highly beneficial for standard FL approaches as well. For example, first pre-training on non-DP clients.

For the second, we analyze robustness to data heterogeneity by using an **alternative** arrangement of `The Pile` which does not contain an inherent cluster relationship. As observed in Table 3 and Figure 4, swapping the data of `WK` with `PBA` harms performance due to the $\mathcal{K}$ layers of the root being unable to agree during attention-based aggregation. Crucially, as can be observed from Fig. 4, the personalized layers of the other nodes, together with the residual mechanism of *WorldLM* allow them to maintain performance despite this decrease for the root—which drives the performance decrease shown in Table 3. Consequently, for a majority of the participating organizations *WorldLM* serves as a superior alternative to FL from a personalization perspective.

### 5.3 *WorldLM* IMPROVES DOWNSTREAM TASK PERFORMANCE

Table 4: Downstream task evaluation on five broad categories evaluated using the MosaicML Gauntlet for models trained on `MC4`. Scores account for the random baseline and average accuracy values within each category uniformly. We report performance relative to the baselines, FL/Centralized, respectively. Full downstream evaluation results are available in Appendix A.6.

| | | | Collaborative | | | | Non-Collaborative | |
|---|---|---|---|---|---|---|---|---|
| Dataset | Model | Category | WorldLM | FL | FedPer | HierFAVG | Local | Centralized |
| **MC4** | **250M** | **comm** | -1.11% | 0.43% | 3.52% | -0.19% | -0.57% | 0.42% |
| **MC4** | **250M** | **lang** | 3.21% | 0.29% | 1.03% | 1.12% | -1.72% | 0.31% |
| **MC4** | **250M** | **read** | 0.43% | 0.21% | -2.14% | 0.12% | -3.49% | 0.21% |
| **MC4** | **250M** | **symbol** | 14.90% | 0.06% | -5.04% | 10.14% | 2.76% | 0.08% |
| **MC4** | **250M** | **know** | 5.04% | 0.21% | 0.21% | 2.85% | 1.19% | 0.22% |
| **Avg Improvement** | | | **4.49%** | | -0.48% | 2.81% | **-0.37%** | |

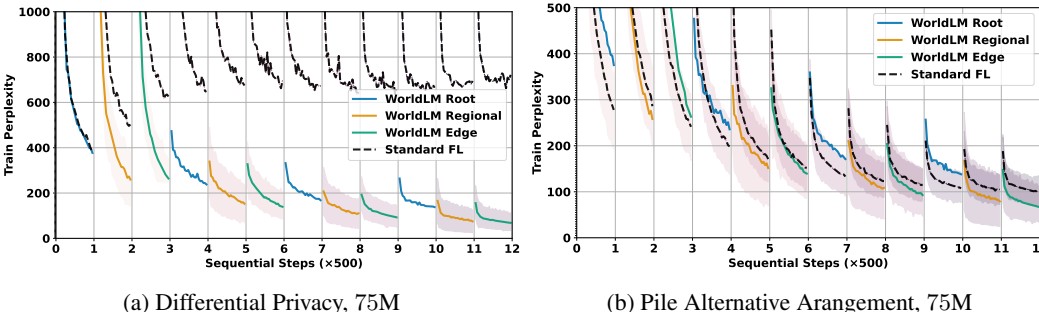

(a) Differential Privacy, 75M    (b) Pile Alternative Arangement, 75M

Figure 4: The impact of differential privacy on *WorldLM* performance, with the 75M model on `The Pile`, versus standard FL. *WorldLM* is able to stabilize the training and validation performance (a,b) when applying DP over the leaf **CC** and **WK** clients, unlike FL which diverges. Similarly, for the alternative arrangement of `The Pile` where `WK` and `PBA` are swapped, the cross-federation sharing mechanism allows *WorldLM* to maintain a good degree of performance.

To further evaluate the efficacy of *WorldLM*, we assess its performance across various downstream tasks provided by the MosaicML Gauntlet, comparing it with other collaborative and non-collaborative methods. Because our gauntlet includes 35 different benchmarks, we report the averages weighted scores reported by the gauntlet with its necessary adjustments for the random baseline accuracy. As can be seen in Table 4, *WorldLM* achieves superior performance to standard FL baseline, and outperforms *FedPer* and *HierFAVG*. We want to emphasize that few-shot evaluations at this scale are difficult Brown et al. (2020) resulting in low absolute performance, and likely noisy. Thus, Table 4 reports performance across all models in relative terms to the relevant baseline as is done for all comparisons in Appendix A.6. Finally, it is important to note that perplexity is highly predictive of downstream tasks Dubey et al. (2024) and should be the preferred metric for comparison.

## 5.4 LIMITATIONS

The primary limitations of our design related to the need for an implicit relationship to exist across the data of clients in the same federation. While our information-sharing and partial-personalization address this by increasing the distance between two nodes that can result in an interaction, allowing *WorldLM* to perform well on the alternative arrangement of `The Pile`, it is not a guaranteed solution. Other significant limitations relate to preventing attacks in a hierarchical scenario, as a single centralized control point does not exist. However, the residuals of *WorldLM* could be used to potentially filter out outliers from aggregation. A full description of the limitations of *WorldLM* is available in Appendix A.4.

## 6 CONCLUSION

*WorldLM* supports the extension of federated learning (FL) to the challenging setting of worldwide optimization of language models (LMs). Our results indicate that systems based on **federations-of-federations** can compete with standard FL and centralized optimization for the medium-sized LMs affordable to small organizations and groups, given their hardware. Our results show that *WorldLM* can outperform standard FL under realistic federated topologies and data distributions constructed using naturally heterogeneous datasets. Furthermore, they also indicate our method to be robust under the constraints of differential privacy, unlike standard FL. Thus, *WorldLM* is an effective approach for addressing the nascent sub-field of worldwide LM pre-training. We open several new research opportunities such as: (a) defending against model poisoning in hierarchical settings, (b) bringing the benefits of *WorldLM* billion-scale models, (c) tackling broader forms of statistical heterogeneity, and (d) applying it to parameter-efficient fine-tuning. We hope this will help democratize LM training across national boundaries and address the societal concerns regarding its governance.

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

# A APPENDIX

Table 5: Hyperparameters for *WorldLM*. The federated learning rate $\eta_s$ and momentum $\mu_s$ (Huo et al., 2020) are applied by a standard FL server or **each** server of *WorldLM*. $|\mathcal{K}|$ is the number of blocks used for the key in *WorldLM*, while $\nu_{\mathcal{K}}$ is the number of layers selected for each residual across all clients. Finally, $\mathbf{S_C}$ represents the parameters of the learning rate scheduler synchronized across **sequential** steps. Do note that the 400M model is trained for twice as many steps per round (Hoffmann et al., 2022).

| Model (size) | #Rounds | $\eta_s$ | $\mu_s$ | $|\mathcal{K}|$ | $\nu_{\mathcal{K}}$ | $\mathbf{S_C}(\alpha,\ \eta_{max},\ \mathbf{T})$ |
|---|---|---|---|---|---|---|
| **English (75M)** | 12 | 0.2 | 0.9 | 1 | 1 | $(10^{-2},\ 8 \times 10^{-4},\ 3 \times 10^3)$ |
| **English (125M)** | 21 | 0.2 | 0.9 | 3 | 1 | $(10^{-2},\ 6 \times 10^{-4},\ 5 \times 10^3)$ |
| **Multi (250M)** | 21 | 0.2 | 0.9 | 1 | 1 | $(10^{-2},\ 8 \times 10^{-4},\ 5 \times 10^3)$ |
| **Multi (400M)** | 21 | 0.2 | 0.9 | 1 | 1 | $(10^{-2},\ 3 \times 10^{-4},\ 1 \times 10^4)$ |

Table 6: Architecture details and local training parameters for our 75M and 250M models. They represent the number of transformer blocks, hidden model dimension, number of attention heads, the linear layer expansion ratio and the parameters of Adam.

| Model (size) | #Blocks | $d$ | #Heads | Exp. Ratio | $(\beta_1,\ \beta_2)$ | $|$Vocab$|$ |
|---|---|---|---|---|---|---|
| **English (75M)** | 3 | 896 | 16 | 4 | (0.9, 0.95) | 50K |
| **English (125M)** | 12 | 768 | 12 | 4 | (0.9, 0.95) | 50K |
| **Multi (250M)** | 3 | 896 | 16 | 4 | (0.9, 0.95) | 250K |
| **Multi (400M)** | 16 | 896 | 16 | 4 | (0.9, 0.95) | 250K |

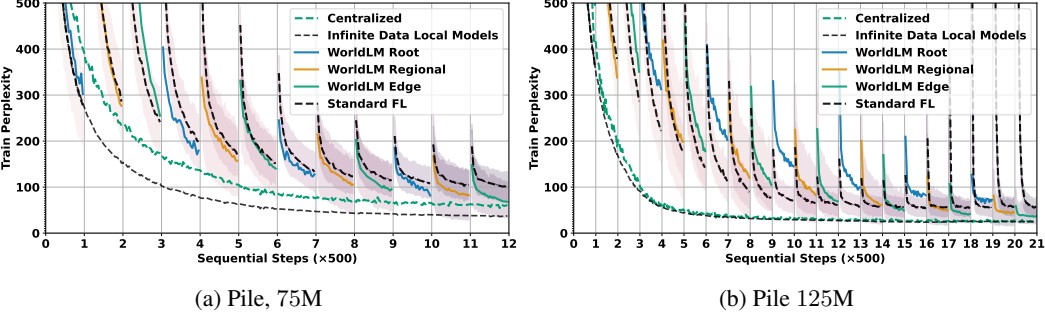

(a) Pile, 75M          (b) Pile 125M

Figure 5: *WorldLM* training and performance of the 75M) and 125 English models on a three-level heterogeneous partitioning of *The Pile* (Fig. 2). While the hierarchical approach makes steady progress due to its attention-based aggregation and partial personalization, standard FL struggles to converge due to data heterogeneity. Crucially, the performance of *WorldLM* approaches that of the centralized model and partially overlaps with infinite-data local models.

## A.1 THE IMPACT OF SAMPLING ON WORLDLM

In the WorldLM algorithm, the size of a data source impacts the sampling ratios within the local training pipeline of a given node and the composition of the evaluation set. The hierarchical structure of the federations necessitates a multi-layered approach to sampling, where each level of the hierarchy influences the effective sampling ratios for the data sources.

### A.1.1 LOCAL SAMPLING STRATEGY

Let $\omega_c$ represent the dataset held by client $c$. If a participant holds a single data source, they exclusively sample from that source. For example, a participant holding only Common Crawl (CC) samples will see as many samples of CC as a participant holding only PubMed Central (PBC). Conversely, if a participant holds both Wikipedia (WK) and Common Crawl (CC) datasets, they sample from these sources proportionally to their sizes when forming batches. The local sampling ratio, $r_{\omega_c}$, for data from a source $\omega_c$ at a given node is thus determined by the relative sizes of the datasets held by that node.

### A.1.2 EFFECTIVE SAMPLING RATIO ACROSS HIERARCHY

To determine the effective sampling ratio for a given data source across all model training steps, one must consider the data mixture across all clients in the federation. It is important to keep the sequential versus parallel distinction in mind. For example, if in standard FL, 7 clients would executed in parallel and be averaged, the execution of *WorldLM* would depend on the hierarchical structure. Thus, while standard FL could complete 3 rounds with 250 steps per client, for a total number of parallel SGD steps of 5250, *WorldLM* would only perform $250 + 250 * 2 + 250 * 4 = 1750$ parallel steps despite doing the same number of sequential steps.

Let us formalize the sampling ratios using a concrete example involving data sources from the Multilingual Colossal Common Crawl (mC4).

Formulation for Effective Sampling Ratio

We define the batch-wise sampling ratio for the Bulgarian dataset, $\omega_{\mathrm{BG}}$, considering all levels of the hierarchy. The sampling ratio $r_{\mathrm{BG}}$ is given by summing contributions from each hierarchical level. This can be expressed as:

$$r_{\mathrm{BG}} = \frac{|\omega_{\mathrm{BG}}|}{|\omega_{\mathrm{FR}}| + |\omega_{\mathrm{IT}}| + |\omega_{\mathrm{BG}}| + |\omega_{\mathrm{UK}}|} + \frac{|\omega_{\mathrm{BG}}|}{|\omega_{\mathrm{UK}}| + |\omega_{\mathrm{BG}}|} + \frac{|\omega_{\mathrm{BG}}|}{|\omega_{\mathrm{BG}}|}$$

where: - The first term represents the root server sampling from all four datasets proportionally to their sizes in each batch. - The second term represents a regional server sampling from UK (UK) and Bulgarian (BG) proportionally to their sizes. - The third term represents a leaf node containing only Bulgarian data.

### A.1.3 IMPACT OF AGGREGATION PROCEDURE

Next, we consider the impact of the aggregation procedure on the sampling ratio. To model the sampling ratio given the effect of averaging, we account for the number of participants at each hierarchical level. The adjusted sampling ratio $\overline{r_{\mathrm{BG}}}$, taking into account hierarchical averaging, is:

$$\overline{r_{\mathrm{BG}}} = \frac{|\omega_{\mathrm{BG}}|}{|\omega_{\mathrm{FR}}| + |\omega_{\mathrm{IT}}| + |\omega_{\mathrm{BG}}| + |\omega_{\mathrm{UK}}|} + \frac{1}{2} \frac{|\omega_{\mathrm{BG}}|}{|\omega_{\mathrm{UK}}| + |\omega_{\mathrm{BG}}|} + \frac{1}{4} \frac{|\omega_{\mathrm{BG}}|}{|\omega_{\mathrm{BG}}|}$$

This adjusted ratio emphasizes that the dataset size becomes less relevant due to the inherent upsampling effect within each leaf node, facilitated by the hierarchical structure of WorldLM.

## A.2 FEDERATION SIZE SIZES

Table 7 provides the detailed sample sizes for each federation partition. Crucially, only the relative size of a given data source within a single node matters. To construct our splits we partition each dataset into equal shared and then divide the shards between the nodes of the federation, giving 3 shards of each data source to each leaf node, and 2 shards for the other two levels.

Table 7: Number of samples for each federation partition in the datasets MC4 and The Pile. The sizes are in GiB, which serves as a proxy for the number of samples, depending on the tokenizer and sequence length.

| Federation | Dataset | Size [GiB] |
|---|---|---|
| **MC4** | | |
| IT | Italian (IT) | 252.2 |
| FR | French (FR) | 483.1 |
| UK | Ukrainian (UK) | 84.5 |
| BG | Bulgarian (BG) | 44.3 |
| IT + FR | Italian (IT) + French (FR) | 168.1 + 332 |
| UK + BG | Ukrainian (UK) + Bulgarian (BG) | 56.3 + 29.3 |
| IT + FR + UK + BG | | **585.7** |
| **The Pile** | | |
| CC | Common Crawl (CC) | 97.2 |
| WK | Wikipedia (WK) | 2.7 |
| PBC | PubMed Central (PBC) | 38.3 |
| PBA | PubMed Abstracts (PBA) | 8.2 |
| CC + WK | Common Crawl (CC) + Wikipedia (WK) | 64.8 + 1.8 |
| PBC + PBA | PubMed Central (PBC) + PubMed Abstracts (PBA) | 25.5 + 5.5 |
| CC + WK + PBC + PBA | | **97.6** |

## A.3 Detailed Algorithm Description

In this appendix, we provide a comprehensive and detailed description of the WorldLM algorithm, aimed at offering a thorough understanding of its operations.

### A.3.1 Algorithm Overview

Each participant in WorldLM is referenced with a node ID $q \in \{0, 1, \ldots, n-1\}$. Every node $q$ possesses a parent $p$, except for the root node $q = 0$, and a set of descendants $C_q$. The following detailed steps outline the execution of the WorldLM algorithm for node $q$.

### A.3.2 Initial Setup

Each node $q$ is initialized with:

- A backbone $B_q^0$ and an ordered sequence of personalized key layers $K_q^0$, composed of pairs $(v, l)$, where $v$ are the parameters of a layer and $l$ is their index in the model.
- Two unordered sequences of downstream residuals: one for pre-training aggregation, $D_a$, and one for downstream routing, $D_r^0$, both consisting of pairs $(v, l)$.
- Federated server optimization method, ServerOpt, and client optimization method, ClientOpt.
- A similarity function, typically cosine similarity.
- Number of training rounds, $T_q$.

### A.3.3 Execution Steps

The algorithm for node $q$ proceeds as follows:

1. **Initialize Parameters:** Node $q$ loads its initial parameters, separating them into the backbone $B_q^0$ and the personalized key layers $K_q^0$.
2. **Aggregate Initial Key Layers:** If $q$ is not the root node:
   - (a) Aggregate $K_q^0$ with downstream residuals for pre-training aggregation, $D_a$, and the parent key layers $K_p$ using a layer-wise attention mechanism.

(b) Expand the pairs in $D_a$ to a complete model, masking unfilled layers. For each element $r \in D_a$, obtain a new ordered sequence $K_r$.

(c) Treat each current node's key layer $K_{q,l}$ as a query $\mathcal{Q}_l$ in a self-attention mechanism involving $K_{q,l}$, $K_{p,l}$, and $K_{r,l}$ for all residuals:

$$e_{q,q} = 1$$

$$e_{q,p} = \frac{\langle K_{q,l}, K_{p,l} \rangle}{|K_{q,l}||K_{p,l}|}$$

$$e_{q,r} = \frac{\langle K_{q,l}, K_{r,l} \rangle}{|K_{q,l}||K_{r,l}|}, \forall r \in D_a$$

(d) Compute attention coefficients $\lambda$ using a softmax over the cosine similarities and produce the final layer at position $l$:

$$K'_{q,l} = \lambda_{q,q} K_{q,l} + \lambda_{q,p} K_{p,l} + \sum_{r=1}^{|D_a|} \lambda_{q,r} K_{r,l}$$

3. **Replace Backbone:** Replace the loaded backbone $B_q^0$ with the parent backbone $B_p$.

4. **Training Round Execution:**

   (a) For each training round $t \in \{0, \dots, T_q - 1\}$:

      i. Train the model of node $q$ using ClientOpt.

      ii. Route each downstream residual layer $(v, l)$ received from the parent $D_r^t$. For a key layer $v$ with index $l$, compute its similarity to $K_{c,l}$ for all children $c \in C_q$:

$$e_c = \langle v, K_{c,l} \rangle, \forall c \in C_q$$

      iii. Select the destination node with the highest cosine similarity. If the child is a leaf, send the layer as part of the residuals to be aggregated $A^t$ or routed $R^t$. All children execute recursively in parallel.

      iv. Aggregate the key layers of descendants. If $q$ is not a leaf node, perform:

         A. Compute pseudo-gradients $\Delta_c^t \leftarrow B_c^t - B^t$ for each child received backbone $B_c^t$.

         B. Average pseudo-gradients into $\Delta^t$ and apply $\Delta^t$ to the backbone using ServerOpt.

         C. Perform full attention computation where the layer of a child $K_{c,l}$ serves as key, value, and query:

$$e_{c,j,l} = \frac{\langle K_{c,l}, K_{j,l} \rangle}{|K_{c,l}||K_{j,l}|}, \forall j \in C_q \cup \{q\}$$

$$\lambda_{c,j,l} = \frac{\exp(e_{c,j,l})}{\sum_{z \in C_q \cup \{q\}} \exp(e_{c,z,l})}$$

$$K'_{c,l} = \sum_{j \in C_q \cup \{q\}} \lambda_{c,j,l} K_{j,l}$$

         D. Average attentional representations:

$$K'_{q,l} = \sum_{c \in C_q \cup \{q\}} \frac{1}{|C_q \cup \{q\}|} K'_{c,l}$$

         E. For each layer position $l$, score the same layer across all clients $K_{c,l}$ against $K'_{q,l}$ and select the $\nu_{\mathcal{K}}$ most dissimilar layers to be sent upstream as residuals or routed down if $q$ is the root.

### A.3.4 MATHEMATICAL REPRESENTATION OF KEY PROCEDURES

The following equations summarize the key procedures involved in the algorithm, making heavy use of attention aggregation and similarity measures.

### A.3.5 Layer-wise Attention Mechanism

Given the query, key, and value representations of key layers:

$$\mathcal{Q}_l = K_{q,l}$$
$$\mathcal{K}_l = \{K_{p,l}, K_{r,l} \forall r \in D_a\}$$
$$\mathcal{V}_l = \{K_{q,l}, K_{p,l}, K_{r,l} \forall r \in D_a\}$$

The similarity scores and attention weights are computed as:

$$e_{q,i,l} = \frac{\langle \mathcal{Q}_l, \mathcal{K}_{i,l} \rangle}{|\mathcal{Q}_l||\mathcal{K}_{i,l}|}$$

$$\lambda_{q,i,l} = \mathrm{softmax}(e_{q,i,l}) = \frac{\exp(e_{q,i,l})}{\sum_j \exp(e_{q,j,l})}$$

The final aggregated key layer is then:

$$K'_{q,l} = \sum_i \lambda_{q,i,l} \mathcal{V}_{i,l}$$

These steps are repeated for each training round and for each layer position $l$ as described above, ensuring that the model aggregates information from various levels of the hierarchy in a structured and efficient manner.

### A.3.6 Pseudo-Gradient Computation and Aggregation

For a given child's received backbone $B_c^t$ and the current node's backbone $B^t$, the pseudo-gradient $\Delta_c^t$ is computed as:

$$\Delta_c^t = B_c^t - B^t$$

The pseudo-gradients from all children are then averaged to obtain the final gradient update $\Delta^t$, which is applied to the backbone using the federated optimization method $\mathrm{ServerOpt}$.

### A.3.7 Residual Routing and Aggregation

The residual layers $(v, l)$ are routed based on their similarity to the key layers of recipients. The similarity is computed as:

$$e_c = \frac{\langle v, K_{c,l} \rangle}{|v||K_{c,l}|}, \forall c \in C_q$$

The residual is then sent to the child with the highest similarity, ensuring relevant updates are propagated through the hierarchy.

## A.4 Limitations

The limitations of WorldLM come from two different sources: (a) those common to all federated methods [1], (b) those induced by its particular design choices such as the hierarchical structure, partially personalized attention-based aggregation and residual information sharing.

The limitations common to all federated methods which are highly relevant to WorldLM are data heterogeneity, system heterogeneity, and sample efficiency. While we have built WorldLM to tackle data heterogeneity, our experimental setup cannot completely replicate the most pathological natural data distributions which can naturally arise since it relies on known and curated datasets. Thus, it is possible for a real-world federation to hold data that is even more heterogeneous than what we can explore using multiple languages or genres of text, as is the case for the work of Charles et al. (2023). Investigating how to model heterogeneity for federated LM pre-training is an active research topic. Stragglers may be caused by system heterogeneity as nodes with less powerful GPUs may take longer to complete a round, our work assumes that the participants are roughly equal in terms of computational ability and that the impact of potential stragglers on training time is limited.

However, we would like to mention that several opportunities to address this concern, including asynchronous execution [3], partial participation[4], and load balancing[5]. Finally, the sample efficiency of federated averaging and its derived methods is questionable, given that the multiple parallel updates of simultaneously executing participants are averaged together which may result in an uninformative pseudo-gradient on the server [6]. Creating more effective aggregation methods is one of the primary pursuits of the field. While the partially-personalized aggregation of WorldLM may help tackle this issue, the challenge is guaranteed for exist in the case of the backbone aggregation.

The WorldLM design itself suffers from challenges when dealing with data heterogeneity, communication requirements, and potential attacks. As discussed in section 3 of the main work, WorldLM assumes that there exists multiple participants in the federation which share similar data and may be connected either via an ancestor or via the residual-sharing mechanism. When this relation fails to hold, the attention aggregation and residual sharing of WorldLM do not provide a direct benefit over standard FL methods, as shown by FL performing similarly to WorldLM on the C4 dataset. When such a relationship is indeed present and the residual mechanism can take care of it, it is possible for communication restrictions to exist between regions for legal or practical reasons, limiting the efficacy of the residual sharing. Finally, defenses against potential poisoning attacks must now consider all levels of the hierarchy rather than having a simple dual system where the server is trusted while clients are not, as in traditional FL. WorldLM provides some recourse to this as the attention coefficients may indicate if the data distribution of a particular participant is completely out of the ordinary, however, security remains an open future direction.

### A.5    THE LEGAL CONTEXT OF LLM TRAINING

The surge in popularity of language models, notably exemplified by the release and widespread adoption of ChatGPT, has accelerated the integration of AI into various sectors. This expansion has subsequently encouraged the development of regulatory frameworks to govern AI technologies. A pioneering effort in this domain is the European Union's Artificial Intelligence Act (the EU AI Act), which represents the first comprehensive legal framework of its kind, anticipated to set a precedent for global AI regulation (Woisetschläger et al., 2024b). The Act encompasses, among various provisions, rigorous data governance guidelines (Art. 10), including adherence to the General Data Protection Regulation (GDPR). This introduces significant challenges for AI developers, particularly concerning the international transfer of data and data de-biasing processes.

*Restrictions in cross-border transfer of data*: The EU's GDPR imposes stringent conditions on the international transfer of personal data, particularly to third countries deemed to lack adequate protection for personal data. The criterion for 'adequacy', as established in Schrems v DPC (C-262/14)) and Recital 104, requires a level of protection 'essentially equivalent' to that of the EU, a high bar for international data transfers, especially to developing countries. Additionally, the requirement for periodic reviews of adequacy for jurisdictions considered equivalent, alongside mandated safeguard measures (Art. 49) for transfers to non-equivalent third countries, introduces a layer of uncertainty and financial burden for businesses engaged in data transfer. Moreover, the EU is not the only jurisdiction tightening controls over data transfer. China, for instance, has enacted laws and supplementary provisions mandating a security assessment by regulator for the transfer of 'important data' or personal data exceeding specific thresholds, barring certain exemptions. The varied landscape of regulations poses challenges for accessing diverse local datasets while adhering to disparate, and sometimes significantly different, regulations across jurisdictions. Hierarchical FL offers an efficient solution to maintain compliance by storing data and model within its jurisdiction of origin, avoiding cross-border transfers.

*Mitigation of data bias*: The EU AI Act mandates rigorous oversight throughout the entire lifecycle of the data used in AI models and obligates the implementation of *'appropriate measures to detect, prevent, and mitigate potential biases'* (Art. 10.2f,fa). It underscores the growing importance of accessing diversified data sources, particularly those from jurisdictions that are underrepresented. To accommodate the acquisition of such data while adhering to individual privacy and local regulations, a novel training paradigm, comprising multiple layers of federation both among clients and jurisdictions, is necessitated to address the limitations of the traditional single-layer FL framework.

*The right to information*: Transparency regarding the data collected and utilized in training, along with the rights of AI developers to access such information, has garnered considerable attention from

both regulators and content creators. The GDPR grants individuals the right to obtain all information stored by a service provider (Art. 15, Rec. 63 & 64), including details on the application of this data in training models. Furthermore, the EU AI Act mandates that model providers compile and disclose a comprehensive summary of the training content publicly (Art. 52c). In the US, legal disputes such as "Times v OpenAI" have underscored the debate over the extent to which the fair use doctrine under US copyright law protects the utilization of copyrighted materials in the training of AI models. This case also ignites broader discussions about the adequacy of the current legal framework in safeguarding content creators against the opaque practices of LLM training. These challenges have led to legislative proposals, including the Generative AI Copyright Disclosure Bill in the US House of Representatives, aimed at enhancing transparency and accountability. Hierarchical FL, by its design, offers inherent advantages over centralized training models by delineating clear data provenance — identifying the sources of data and their contributors. This attribute of Hierarchical FL positions it favorably in addressing concerns related to informational rights and data privacy, presenting a more transparent framework for data utilization in AI development.

*Energy efficiency*: The EU AI Act advocates for the environmentally sustainable development of AI systems by proposing the formulation of a Code of Conduct. This Code is intended to establish explicit objectives and key performance indicators (Art. 69), mirroring the core values of the EU but also responding to growing concerns within the industry and broader society regarding the energy consumption associated with the training and use of AI.

## A.6 Downstream Task Analysis

We report values on the gauntlet provided by MosaicML, using the full gauntlet with category definitions and random baseline accuracy as provided by Mosaic.

| | | | Collaborative | | | | Non-Collaborative | |
|---|---|---|---|---|---|---|---|---|
| Dataset | Model | Category | WorldLM | FL | FedPer | HierFAVG | Local | Centralized |
| MC4 | 250M | comm | -1.11% | 0.43% | 3.52% | -0.19% | -0.57% | 0.42% |
| MC4 | 250M | lang | 3.21% | 0.29% | 1.03% | 1.12% | -1.72% | 0.31% |
| MC4 | 250M | read | 0.43% | 0.21% | -2.14% | 0.12% | -3.49% | 0.21% |
| MC4 | 250M | symbol | 14.90% | 0.06% | -5.04% | 10.14% | 2.76% | 0.08% |
| MC4 | 250M | know | 5.04% | 0.21% | 0.21% | 2.85% | 1.19% | 0.22% |
| **Avg Improvement** | | | **4.49%** | | -0.48% | 2.81% | **-0.37%** | |

| Dataset | Model | Category | Collaborative | | Non-Collaborative | |
|---------|-------|----------|---------------|------|-------------------|-------------|
| | | | WorldLM | FL | Local | Centralized |
| Pile | 125M | comm | 2.79% | 0.40% | 4.74% | 0.40% |
| Pile | 75M | comm | 0.26% | 0.42% | -0.79% | 0.41% |
| Pile | 125M | lang | -0.27% | 0.31% | -3.15% | 0.33% |
| Pile | 75M | lang | -1.89% | 0.31% | -0.81% | 0.32% |
| Pile | 125M | read | 1.88% | 0.20% | 11.43% | 0.24% |
| Pile | 75M | read | -0.15% | 0.21% | 9.67% | 0.20% |
| Pile | 125M | symbol | 2.23% | 0.08% | 5.89% | 0.08% |
| Pile | 75M | symbol | 7.91% | 0.07% | 6.02% | 0.08% |
| Pile | 125M | know | -3.84% | 0.22% | -1.75% | 0.23% |
| Pile | 75M | know | 0.86% | 0.21% | 2.20% | 0.21% |
| **Avg Improvement** | | | **1.40%** | | **0.32%** | |

| Dataset | Model | Category | Collaborative | | Non-Collaborative | |
|---------|-------|----------|---------------|------|-------------------|-------------|
| | | | WorldLM | FL | Local | Centralized |
| C4 | 75M | comm | -2.59% | 0.42% | - | 0.40% |
| C4 | 75M | lang | -1.57% | 0.30% | - | 0.32% |
| C4 | 75M | read | 4.09% | 0.21% | - | 0.23% |
| C4 | 75M | symbol | -1.36% | 0.08% | - | 0.08% |
| C4 | 75M | know | 0.03% | 0.20% | - | 0.22% |
| **Avg Improvement** | | | **-0.28%** | | | |

| Dataset | Model | Category | Collaborative | |
|------------|-------|----------|---------------|------|
| | | | WorldLM | FL |
| Pile (Alt) | 75M | comm | 2.79% | 0.42% |
| Pile (Alt) | 75M | lang | 0.26% | 0.31% |
| Pile (Alt) | 75M | read | -0.27% | 0.21% |
| Pile (Alt) | 75M | symbol | -1.89% | 0.07% |
| Pile (Alt) | 75M | know | 1.88% | 0.21% |
| **Avg Improvement** | | | **1.40%** | |

