# OpenReview forum: "Worldwide Federated Training of Language Models"
_ICLR.cc/2025/Conference — ICLR 2025 Conference Withdrawn Submission_

### Official Review · Reviewer_nYDU · 2024-11-02

**Soundness:** 2
**Presentation:** 3
**Contribution:** 2
**Rating:** 5
**Confidence:** 3

**Summary:**

The paper proposes WorldLM, a federated language model training system that leverages a "federation of federations" structure. This approach enables collaboration among organizations across different legal, security, and privacy jurisdictions to train language models on heterogeneous data without sharing sensitive information. WorldLM uses a partially personalized aggregation technique and cross-federation information sharing via residual layer embeddings, which addresses statistical heterogeneity in the data. Evaluation on diverse datasets shows that WorldLM outperforms traditional federated learning, closely matches the performance of localized models, and remains effective under differential privacy conditions.

**Strengths:**

1. The federations-of-federations approach allows for adaptable collaboration across various jurisdictions, making it feasible to integrate global, region-specific, or industry-specific data in a way that respects privacy constraints.

2. The backbone with personalized key layers effectively captures and adapts to local variations in data, enhancing performance in heterogeneous settings.

3. WorldLM is robust in applying differential privacy, even where traditional federated learning might struggle, which is a critical advantage for handling sensitive information.

**Weaknesses:**

1. The method shows diminished effectiveness when data within a federation lacks inherent similarity, suggesting a need for improved aggregation techniques for highly diverse datasets.

2. While WorldLM works well on medium-sized language models, scaling to larger models could be resource-intensive, especially for smaller organizations with limited computational resources.

**Questions:**

1. How scalable is WorldLM for much larger models or significantly larger numbers of federations?
2. How does WorldLM handle emerging data distributions in dynamic environments?
3. What specific legal and privacy frameworks were considered in the evaluation?

---

### Official Review · Reviewer_7Yxh · 2024-11-03

**Soundness:** 3
**Presentation:** 2
**Contribution:** 3
**Rating:** 5
**Confidence:** 2

**Summary:**

This paper presents WorldLM, a hierarchical federated learning approach for training language models across organizations, each with its own domain data. The organizations are assumed to have a hierarchy of similar datasets, and it is an appealing idea to exploit this structure in the parameter updates. The key contributions are:
1. a "federation of federations" architecture that allows organizations to collaborate while managing data heterogeneity;
 2. a model decomposition into shared backbone and personalized key layers with attention-based aggregation mechanism to address distribution mismatch across federations;
3. empirical demonstration that this approach outperforms standard federated learning on multilingual and domain-specific datasets, while maintaining strong performance under differential privacy constraints.

While the work addresses an important problem and presents some interesting approaches, there are several significant limitations in the presentation, justification of design choices, and evaluation methodology.

**Strengths:**

* The paper addresses an important practical problem in distributed LM training.
* The attention-based aggregation mechanism is an interesting approach to handling heterogeneous data (though it does not seem to be a complete solution, it suggests an interesting direction).
* Experimental design and evaluation strategy:
  * Evaluation across multiple model sizes (75M-400M parameters). Scaling experiments show competitive performance with standard FL.
  * Comprehensive testing combining perplexity metrics with LLM benchmarks (though these can be reported more clearly)
  * Demonstrates robust performance under differential privacy constraints, outperforming standard FL approaches in these settings.

**Weaknesses:**

1. Presentation and Motivation:
The paper's introduction and related work sections attempt to cover both technical and policy aspects of federated learning, but in doing so, fail to provide a clear technical foundation. While the data regulation context is interesting to learn about, it comes at the expense of a precise technical exposition. At times, the paper mentions low-level technical concepts (e.g., RingAllReduce, local SGD) without proper explanation. The presentation would benefit from a clearer explanation of key technical concepts, such as a precise definition of a federation or “federation of federations”, and a discussion of FL aggregation methods.

The transition from presenting challenges to proposing a "federation of federations" solution lacks sufficient justification. While the abstract and introduction emphasize federated governance and legal/privacy challenges across organizations, it’s unclear how the proposed hierarchical setup addresses these challenges. The gaps between the motivation, formal problem setup (Section 3.1), and algorithm details (Section 3.2) is not fully justified. The term "Worldwide" seems unnecessarily broad for a technical ML paper.

One of the questions addressed by this paper is how we can update a model’s parameters from gradients computed in different tasks, without much interference. This question is heavily studied in multi-task learning, where a single neural network is trained on a collection of different tasks (e.g., see these two influential papers https://arxiv.org/abs/2001.06782, and https://arxiv.org/abs/1705.07115). Could the authors discuss how the proposed approach relates to/differs from these multi-task learning techniques? It seems like the attention mechanism proposed by the authors attempts to achieve similar objectives, but, as I write below, it is unclear whether it leads to consistent improvements because interference seems to be high (Fig 3).

2. Technical Framework:
The mathematical presentation of hierarchical data distributions, while formal, adds complexity without clear benefit. The use of terms like "LDA" with values 0.0001 and 1000 (lines 189-190) is unclear (is this Latent Dirichlet Allocation?). Could the authors provide concrete examples of how their abstract mathematical formulations (Section 3.1) translate to real-world data scenarios?

3. Algorithm Design:
Several critical design choices lack proper justification:
- Why is simple averaging sufficient for the backbone while key layers require attention-based aggregation?
- Why was this specific proportion of backbone vs. key layers chosen (I noticed the 30% explanation of transfer learning, but I wasn’t convinced by its importance).
- What are the "meta-learning properties of FL" that motivate these choices?
- Why is sequential training across levels (root → regional → edge) necessary rather than parallel training?
- Why did the authors choose this attention mechanism and do the results indicate that it’s not resolving interference in the weight updates?

4. Evaluation and Results:
The experimental results raise several concerns:
- Figure 3 shows large perplexity spikes between levels, suggesting potential instability, but the implications aren't discussed. Could the authors provide an explanation for these perplexity spikes and discuss how they affect the stability and performance of the model?
- Table 3 / Figure 4 seem to suggest the proposed method is in fact not very robust to statistical heterogeneity. Restructuring the federation hierarchy to combine non-heterogeneous data results in almost 2x increase in perplexity scores. More detailed experiments regarding the method’s robustness could be insightful. For example, instead of entirely swapping the WK and PBA datasets in the robustness experiment (Fig 2), could the authors show the results as X% of WK and PBA are swapped, varying X from 0 to 100?
- It's unclear whether perplexity is measured on consistent validation sets across levels?
- I understand that the models here are not on the multi-billion parameter scale, yet, the high perplexity values (in the hundreds) and large fluctuations seem somewhat too high for a modern LM. Could the authors provide some context for these perplexity values (e.g. comparisons to models of similar size) or discuss why these values are reasonable given their experimental setup?
- Lack of ablations for robustness to DP results: it’s unclear which aspect of the proposed method (hierarchical modeling, algorithm details, attention aggregation method?) is responsible for the robustness compared to standard DP.
- The MosaicML gauntlet results are presented as percentage improvements without raw scores, making it difficult to assess their significance given the known variance in these benchmarks (unless I am missing something).
- The sequential training approach introduces practical deployment challenges that aren't addressed

5. Practical Limitations:
The sequential training across levels appears to be a major limitation that isn't thoroughly discussed. Questions about deployment readiness, training efficiency, and stability during level transitions remain unanswered.

**Questions:**

- Why is simple averaging sufficient for the backbone while key layers require attention-based aggregation?
- Why was this specific proportion of backbone vs. key layers chosen (I noticed the 30% explanation of transfer learning)
- What are the "meta-learning properties of FL" that motivate these choices?
- Why is sequential training across levels (root → regional → edge) necessary rather than parallel training?
- Why did the authors choose this attention mechanism and do the results indicate that it’s not resolving interference in the weight updates?

---

### Official Review · Reviewer_oV6s · 2024-11-04

**Soundness:** 2
**Presentation:** 2
**Contribution:** 2
**Rating:** 3
**Confidence:** 4

**Summary:**

In this paper, the authors focused on a hierarchical way of training federated models.
The key idea is to use attention-based aggregation and residual embedding sharing
to enable learning over "federations of federations" efficently.

**Strengths:**

- The idea seems to make sense, although its practical application could be better motivated

**Weaknesses:**

- I would appreciate if the authors can motivate the problem more concretely, rather than
in a high-level way
- I am a little bit confused by the comparison and the key insights we can get from these
results. (see Questions)

**Questions:**

I found the comparison against FL and Centralized a little bit confusing and it
would be great if the authors can elaborate more:

1. I don't get why 400M WorldLM can be better than Centralized, and would
appreciate some explanation here. Is it because WorldLM personalized further
more against the global data?

If this is the case, this paper should compare with more personalization-based
method? Even training a lora adapter for each person seems to be a stronger
method?

2. " For the 400M, we only had the resources for comparison against centralized,
as standard FL is much less computationally efficient" -- I am not sure I get this --
simulate FL should be cheaper than simulating the centralized setting?

---

### Official Review · Reviewer_QgAV · 2024-11-04

**Soundness:** 2
**Presentation:** 2
**Contribution:** 2
**Rating:** 3
**Confidence:** 4

**Summary:**

In this submission, the authors present WorldLM, a method that can enable collaborations among different organizations/data owners to train language model.

**Strengths:**

1. To address the data heterogeneity, authors propose attention-based aggregation and residual embeddings, which are very suitable for language model training.

2. The experiments are well-designed, different aspects (such as FL v.s. centralized, privacy, local task performance) are covered. These results are promising.

**Weaknesses:**

1) Scale can be large.
It is great that the authors proposed specific-designed FL method for language model (LM). As we know, LM becomes useful when the scale is very large, i.e., LLM. Thus whether the proposed method can be scaled up is really important. Currently, the authors conduct experiments with a largest size of 400M parameters, which is still small. As the experiments with large size might be difficult, could the authors discuss more details when the size of LM reaches billion scale, given their experiments with 75M ~ 400M? For example, discuss potential challenges in communication, memory requirements, or convergence behavior as the model size increases to billion-scale. Additionally,  how the attention-based aggregation and residual embedding mechanisms might behave at larger scales.

2) Communication should be discussed and experimentally tested.
As the presented WorldLM aims collaborations among different data owners, it is necessary to analyze and further experimentally test the communication costs, at lease for the designed simulations, i.e., (a) and (b) in the first paragraph of section 4. It would be better to have quantitative comparisons of communication between WorldLM and other federated learning approaches. Additionally, it would be helpful to discuss how communication costs scale with model size and number of participants in the federation.


3) Technique novelty can be clearly stated.
In section 2.2, related work about PFL, client clustering and hierarchical systems are simply discussed. It is better to clearly state the technique novelty of the proposed method compared to existing methods. In this way, readers can better understand the technique contribution of the proposed method in the field of FL.
For example, authors an provide a concise summary table or paragraph that explicitly compares WorldLM's key features (e.g., attention-based aggregation, residual embeddings) with those of existing methods like PFL, client clustering approaches, and hierarchical systems. This would help readers quickly grasp the unique contributions of WorldLM.

**Questions:**

Please see above Weaknesses.

---

### Official Review · Reviewer_V6vj · 2024-11-05

**Soundness:** 1
**Presentation:** 2
**Contribution:** 2
**Rating:** 3
**Confidence:** 2

**Summary:**

This paper proposes that WorldLM addresses hierarchical federated learning and creates federations of federations, where different federations enforce various regulations in terms of data protection and other competitive constraints. The work conducts extensive experiments to show the effectiveness of the proposed work.

**Strengths:**

1. This work conducts extensive experiments to show its effectiveness.

**Weaknesses:**

1. Section 3.1 mentions that the children under root $q$ has much more critical data heterogeneity than those under parent $p$. I think the authors should justify that this really happens. In my opinion, the setting in clustered FL usually assumes that the clients within a cluster have very similar data distribution while quite distinct from those not in the same cluster [1]. However, in this case, clients may not have such information, i.e., they don't know the data distribution with each other. Besides, I believe the assumption is very strong in hierarchical FL [2]. For example, in terms of data distribution, the heterogeneity between a local orthopedics hospital and a local ophthalmology hospital may be much more severe than the heterogeneity between two orthopedics hospitals in two different cities.
2. The presentation of the algorithm can be improved. One example is the residual layers. After going through the details of the paper, do you mean the personalized layers (decoders/encoders if the model is a transformer)? The authors should put more information on the design of a model.
3. Table 1 and Table 2 show that "local" under "Non-collaborate" performs much better than World LM. I cannot find the details about this baseline. In common practice, it should use local data to train a client's exclusive model. If that is how you implement the baseline, I am curious about the effectiveness of the proposed work.
4. Figure 1 should explicitly mention the meaning of different notions. I had no idea of the figure until I dived into the details of Section 3.

**References:**
[1] Structured Federated Learning through Clustered Additive Modeling
[2] Hierarchical Federated Learning with Multi-Timescale Gradient Correction

**Questions:**

See **Weaknesses**.

---

### Note · Authors · 2024-11-13

**Comment:**

We thank the reviewers for their constructive feedback and will take it into account for any future submissions.

**Withdrawal Confirmation:**

I have read and agree with the venue's withdrawal policy on behalf of myself and my co-authors.